# Evolution of the quantum Hall bulk spectrum into chiral edge states

T. Patlatiuk[1], C.P. Scheller[1], D. Hill[2], Y. Tserkovnyak[2], G. Barak[3], A. Yacoby[3], L.N. Pfeiffer[4], K.W. West[4] & D.M. Zumbühl[1]

One of the most intriguing and fundamental properties of topological systems is the correspondence between the conducting edge states and the gapped bulk spectrum. Here, we use a GaAs cleaved edge quantum wire to perform momentum-resolved spectroscopy of the quantum Hall edge states in a tunnel-coupled 2D electron gas. This reveals the momentum and position of the edge states with unprecedented precision and shows the evolution from very low magnetic fields all the way to high fields where depopulation occurs. We present consistent analytical and numerical models, inferring the edge states from the well-known bulk spectrum, finding excellent agreement with the experiment—thus providing direct evidence for the bulk to edge correspondence. In addition, we observe various features beyond the single-particle picture, such as Fermi level pinning, exchange-enhanced spin splitting and signatures of edge-state reconstruction.

[1] Departement Physik, University of Basel, Klingelbergstrasse 82, CH-4056 Basel, Switzerland. [2] Department of Physics and Astronomy, University of California, Los Angeles, CA 90095, USA. [3] Department of Physics, Harvard University, Cambridge, MA 02138, USA. [4] Department of Electrical Engineering, Princeton University, Princeton, NJ 08544, USA. These authors contributed equally: T. Patlatiuk, C. P. Scheller. Correspondence and requests for materials should be addressed to D.M.Z. (email: dominik.zumbuhl@unibas.ch)

Systems with topologically protected surface states, such as the quantum spin Hall insulator[1,2] and many other topological insulators, are currently attracting great interest. Among the topological states, the integer quantum Hall effect[3] stands out since it was first discovered. It is the most simple case out of which others have emerged, and thus serves as a paradigmatic system. Accessing the surface states in a topological system separately and independently, however, has proven to be challenging for a number of reasons, including disorder, insufficient resolution, or remnant bulk conductivity contaminating transport experiments. Local probes, such as scanning single electron transistors, could in principle overcome the bulk conductivity problem and have been intensely investigated in the context of quantum Hall systems[4–13]. However, moderate spatial resolution and the requirement of large magnetic fields for discriminating among individual edge states have limited existing experiments to low filling factors and prevented tracking the evolution of the quantum hall edges all the way down to low fields.

Previously, tunneling spectroscopy of cleaved edge overgrowth wires has established the system as one of the best realizations of a 1D ballistic conductor, exhibiting distinct signatures such as quantized conductance[14], spin-charge separation[15], charge fractionalization[16], and indication of helical nuclear order induced by the strongly interacting electrons[17,18]. Here, we use a vector magnet to independently control two orthogonal magnetic fields: one to form quantum Hall edge states and another to perform tunneling spectroscopy.

In this work, we use momentum-resolved tunneling spectroscopy to track the guiding center (GC) positions of the quantum Hall edge states with nanometer precision. Over the magnetic field evolution, we observe first magnetic compression towards the sample edge, and then, at higher fields, motion into the bulk and magnetic depopulation of Landau levels (LLs). Note that in this work we are studying integer quantum Hall edge states and not the spin Hall effect or any other topological state. However,

this technique is also applicable to the latter states. Using both an analytical model and numerical solutions for the evolution of edge states in the limit of hard wall confinement[19–21], we are able to match very well the tunneling spectroscopy fingerprint of the conducting edge states from the topologically gapped bulk phase and hence reveal their direct correspondence. Individual edge modes[22–24] are discernible down to unprecedented low magnetic fields $B_z \approx 10$ mT, where the bulk filling factor $\nu$ is about 500. Furthermore, we observe the chiral nature of edge states, as well as Fermi level pinning effects. In addition, interactions lead to signatures of edge reconstruction and exchange-enhanced spin-splitting at large in-plane magnetic fields. We emphasize that this spectroscopy is done at zero bias, thus eliminating heating or lifetime effects.

## Results

**Integer quantum Hall edge states for the hard wall confinement.** A magnetic field $B_z$, applied perpendicular to a 2D electron gas (2DEG), quenches the kinetic energy of free electrons and condenses them into discrete LLs that are energetically separated by the cyclotron energy $\hbar\omega_c$. Here, $\omega_c = eB_z/m^*$ denotes the cyclotron frequency, $e$ the elementary charge, $\hbar$ the reduced Planck constant, and $m^*$ the effective electron mass. Upon approaching the sample edge the electrostatic confinement potential lifts LLs in energy and causes them to intersect with the Fermi energy, thereby forming a corresponding edge state for each bulk LL, see Fig. 1a and Supplementary Fig. 4. Here, we use the Landau gauge (vector potential $A = 0$ at the edge), where the momentum $k_x$ along the quantum wires is a good quantum number that fully characterizes each state. Given $k_x$, all other quantities may be calculated, such as the wave function center of mass (CM) as well as the GC position $Y = k_x l_B^2$, where $l_B = \sqrt{\hbar/(eB_z)}$ denotes the magnetic length. Throughout the paper, the filling factor is defined as $\nu = 2n + g$, where $n = 0, 1, 2, \ldots$ is the orbital Landau level index, and $0 \leq g < 2$ is the spin occupancy.

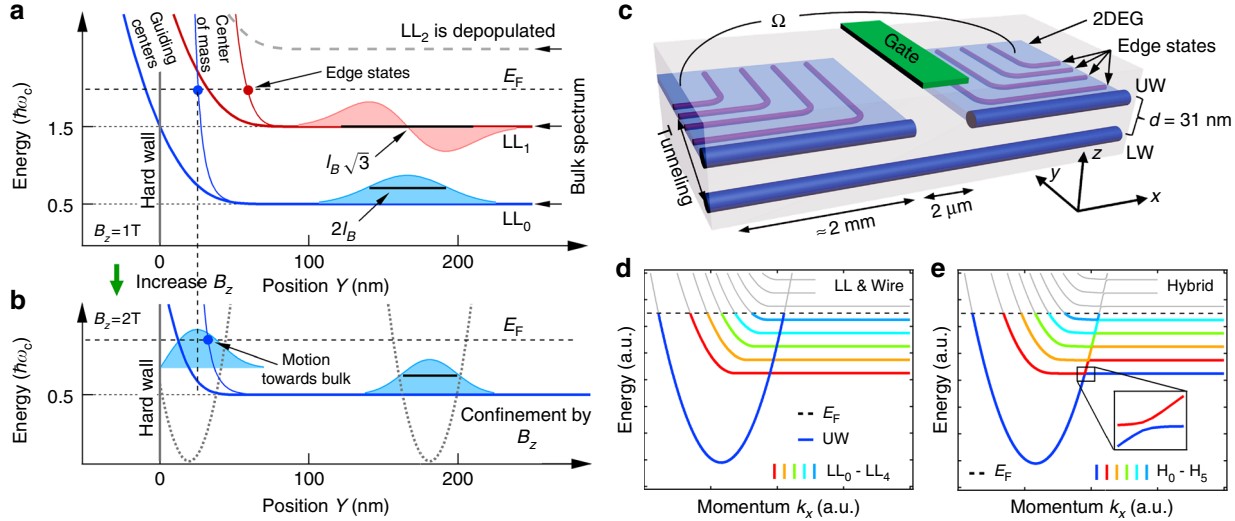

**Fig. 1** Bulk to edge correspondence. **a** Energy evolution of the center of mass (CM) position (thin blue/red curves) and the guiding center (GC) position (bold blue/red curves) for the first two Landau levels $LL_0$ and $LL_1$. Here, $LL_2$ is above the Fermi energy and is depopulated. The confinement (hard wall) lifts the bulk LLs in energy, resulting in corresponding edge states (solid circles) when the CM is crossing the Fermi energy $E_F$. Note that the Fermi energy shown here is lower than in the experiment. **b** Same as **a** for larger magnetic field $B_z$, where the width $2\sigma_n = 2l_B\sqrt{2n+1}$ of $LL_n$ is squeezed and $LL_1$ was depopulated, with magnetic length $l_B$. **c** Coordinate system (black) and sample schematic, showing the 2DEG in light blue, upper and lower quantum wire (UW/LW) in dark blue, top gate in green, and CM for integer quantum Hall edge states in purple. $\Omega$ indicates the conductance measurement. **d** Simplified UW and LL dispersions calculated for independent triangular and hard wall confinement. **e** Calculated dispersion for the combined confinement potential, resulting in hybridized states $H_n$ with avoided crossings (see inset). Here, the bulk $LL_0$ transforms into the UW mode at the sample edge. Gray segments indicate empty states, colored ones are filled

At elevated $B_z$, shown in Fig. 1b, the cyclotron splitting is enhanced. As a consequence the highest LLs are energetically lifted above the Fermi energy and thus magnetically depopulated of electrons, compare Fig. 1a and b. In addition, increasing $B_z$ reduces the magnetic length, thereby squeezing the remaining LL wave functions by magnetic compression and moving the corresponding edge states closer to the sample edge. This holds up to a certain point, when the edge state starts suddenly moving back into the bulk just before being magnetically depopulated, see Fig. 1b.

A sample schematic is depicted in Fig. 1c and consists of two parallel GaAs quantum wells, separated by a thin AlGaAs tunnel barrier. The upper quantum well hosts a high mobility 2DEG, while the bulk of the lower quantum well remains unpopulated. Cleavage of the sample and subsequent overgrowth results in strongly confined 1D-channels in both quantum wells (see Methods section and refs. [14–16,25–31] for more details), termed upper wire (UW) and lower wire (LW) in the following. The LW is used as a tunnel probe to spectroscopically image the integer quantum Hall edge states of the upper quantum well at effectively zero bias voltage.

A simplified picture for the complete dispersion for the upper system is shown in Fig. 1d. It consists of a single localized wire mode UW (dark blue), resulting from the triangular confinement at the sample edge, and the LL spectrum in the presence of hard wall confinement and perpendicular magnetic field $B_z$. Solving the combined electrostatic problem (hard wall confinement with triangular potential near the edge) hybridizes the LL spectrum and quantum wire modes at commensurate conditions where energy and momentum are matched, see Fig. 1e. As a consequence the bulk $LL_0$ transforms into the lowest quantum wire mode at the sample edge. While each LL like edge state in Fig. 1e acquires an additional node in the wave function in comparison to the hard wall spectrum of Fig. 1d, the intersection with the Fermi energy $E_F$ is hardly changed[32,33], giving almost the same effective momentum $k_x$. Therefore, the simplified dispersion of Fig. 1d is used in the following to describe the magnetic field evolution of edge states.

The tunneling regime is obtained by setting the top gate in Fig. 1c to deplete the 2DEG and all UW modes beneath it while preserving a single conducting mode in the LW. This divides the upper system electrically into two halves but preserves tunnel coupling on each side to the LW. Due to translational invariance of UW and LW (away from the top gate, where tunneling occurs), momentum is conserved during the tunneling event, and can be controlled by means of the Lorentz force. In particular, in the presence of an in-plane magnetic field $B_y$ applied perpendicular to the plane spanned by the two wires (see coordinate system in Fig. 1c), tunneling electrons experience a momentum kick $\Delta k_x = -edB_y/\hbar$ along the $x$-direction of free propagation, thus effectively shifting the wire dispersions with respect to each other[15,16,28,29]. Here, $d$ denotes the tunneling distance along the $z$-direction. The resulting zero-bias tunneling conductance is large whenever Fermi points of upper and lower system coincide, see also Supplementary Fig. 1a. In a similar fashion, each LW mode can also be brought into resonance with any given LL. However, in contrast to the quantum wires, the effective momentum of edge modes $k_{x,LL_n}$ of the LLs depends on $B_z$.

**Formation and evolution of the edge states.** Figure 2 shows the measured differential tunneling conductance as a function of magnetic fields $B_z$ and $B_y$. Two horizontal features are visible that correspond to resonant tunneling (energy and momentum

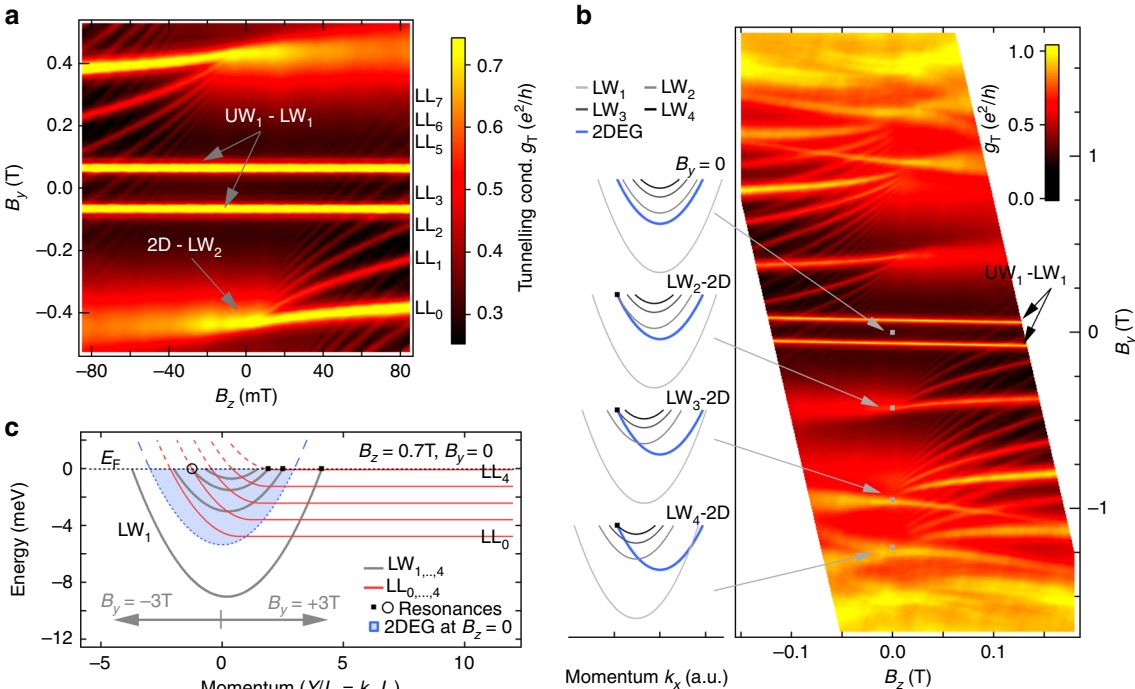

**Fig. 2** Formation and B-field evolution of the chiral integer quantum Hall edge states. **a** Differential tunneling conductance as a function of in-plane magnetic field $B_y$ and perpendicular magnetic field $B_z$ at ≈10 mK. The 2D-wire transitions break up into multiple curves and fan out with increasing $|B_z|$, see also Supplementary Fig. 2. Horizontal resonances at small $|B_y|$, associated with wire-wire tunneling, are not affected by $B_z$. **b** Larger $B_y$ range than **a** showing 6 fans corresponding to tunneling to modes $LW_2$, $LW_3$, and $LW_4$. Due to the chiral nature of the states, the fans are not seen in the data when only $B_z$ is reversed. The sketches depict the resonance condition (black dots) at $B_z = 0$. **c** Schematic representation of quantum wire (gray) to LL (red) tunneling at $B_z = 0.7$ T. $B_y$ shifts the lower wire dispersion in relation to the LLs, as indicated by gray arrows of the corresponding length. The blue filled parabola indicates the 2DEG dispersion for $B_z = 0$, projected onto the $k_x$ axis. Black dots and black circle indicate resonant tunneling to bulk states and edge state, respectively

conservation) between co-propagating electrons of the first upper and lower wire mode, UW$_1$ and LW$_1$, respectively. Since the electron density in UW$_1$ and LW$_1$ is very similar, only little momentum transfer and correspondingly small $\left|B_y\right|$ is required to bring the modes into resonance. These resonances are independent of $B_z$ because the $Y$ coordinates of both UW$_1$ and LW$_1$ modes are very similar. In addition to wire–wire tunneling, extensively studied in the past[15,16,28–31], sharp tunneling resonances of a different origin[22–24] are observed that split into fans of discrete curves in the presence of a perpendicular field and separate with increasing field strength. For each fan, about 10 curves can be resolved down to $B_z \gtrsim 10$ mT, see Fig. 2. As we will show, these fans correspond to resonant tunneling between quantum Hall edge states and the LW modes which are acting as a momentum selective spectrometer. The fan structures observed here track the momentum evolution of edge states with $B_z$ and thereby produce a fingerprint of the conducting edge states. This is in contrast to the conventional Landau fan that simply is an expression of the bulk filling factor as a function of 2DEG density and $B_z$.

The wire modes are supporting states propagating in both negative and positive $x$-direction, irrespective of the perpendicular field, and give transitions extending over both positive and negative $B_z$, see also Supplementary Fig. 1. The quantum Hall edge states, on the other hand, are chiral and are thus propagating only in one direction for a chosen sign of $B_z$ along a given edge. The corresponding LL dispersions are therefore not symmetric under reversal of $k_x$, and the fan structures become directional. Indeed, the fans are seen only for one sign of $B_z$ around a given $B_y$, e.g. in the lower right in Fig. 2a but not the lower left. The opposite sign of $B_z$ also supports a fan but only when $B_y$ is inverted at the same time, i.e., when the total $B$-field is switching sign (Onsager's reciprocity[34]), see upper left in Fig. 2a. This directly indicates the chiral nature of these edge states.

Besides fan structures in Fig. 2a, there are additional fans originating at different $B_y$ values, see Fig. 2b where a larger field range is displayed. These other fans result from tunneling to other modes of the LW. Since each of the different modes in the LW has a different density and thus a different Fermi momentum, an overall momentum shift results, which displaces the fans along $B_y$, as illustrated in the sketches of Fig. 2b, where the resonance condition at $B_z = 0$ is shown (origin of the fans).

In order to quantitatively understand the field evolution of the fan structures, the LL dispersions have to be considered, which depend on the electrostatics at the edge[6,35–41]. For the present samples, the cleavage exposes an atomically sharp edge, which is immediately overgrown by means of lattice matched molecular beam epitaxy[25,26]. The resulting hard wall confinement potential gives rise to the LL dispersions of Fig. 2c (red)[19,21], shown along with the quantum wire modes in the lower well (gray) and the 2DEG at $B_z = 0$ (blue). Lowering $B_z$ reduces the bulk LL energy splitting $\hbar\omega_c$ and hence introduces a more dense LL structure while leaving the LW modes unaffected (due to their strong transverse confinement). The in-plane magnetic field $B_y$, on the other hand, is assumed to not directly affect the LLs, but only shift their dispersion in relation to the LW.

The fan structures at positive $B_Z$ in Fig. 2b can then be understood in terms of momentum-conserving edge state tunneling using one of the LW left Fermi points as a spectrometer, see Fig. 2c, where the case of LW$_3$ is marked with an open circle. Thus, each fan represents a map of the momenta of the LL edge states at the Fermi energy, and thus, via the GC-momentum relation $Y = k_x l_B^2$, a precise map of the GC positions of the edge states. Upon approaching $B_z = 0$, the effective momentum of edge states, i.e. the intersection of LLs with the chemical potential,

approaches the $B_z = 0$ Fermi wave vector $-k_{F,2D}$ of the 2DEG. During this process edge states associated with LLs of increasing orbital index subsequently come into co-propagating resonance with LW$_2$ at $B_y = 0$.

In the following section, the range in the perpendicular magnetic field is extended (Fig. 3) in order to study the field evolution of edge states and their magnetic depopulation at large $B_z$. For better visibility, we plot the second derivative with respect to $B_y$ of the differential tunnel conductance in Fig. 3a. A large number of interpenetrating resonances are visible, extracted in Fig. 3b for clarity, and grouped into bundles according to their different origin, i.e. red, black, and light blue data indicate co-propagating tunneling to the first three LW modes, LW$_1$, LW$_2$, and LW$_3$, respectively. The LL edge states can also be mapped in a counter-propagating fashion (Fig. 3c), i.e. where the wire state and edge states are propagating in opposite directions. To achieve momentum conservation in this case, a relatively large momentum kick needs to be provided by the magnetic field, and these transitions thus appear at larger $B_y$.

In addition to edge state–wire tunneling, intra−wire transitions are seen and color coded in gray in Fig. 3b. As the wave functions for LW$_1$ and UW$_1$ are very similar, their CM positions nearly sit on top of each other and hence there is no resulting momentum kick $\Delta k_x = e\Delta y B_z/\hbar$ due to the perpendicular field. Here, $\Delta y$ denotes their lateral displacement. As a consequence, the corresponding resonances (light gray) appear as horizontal lines. In contrast to this, transitions involving different wire modes, e.g. UW$_1$ and LW$_2$ (dark gray data in Fig. 3b), appear with a slope that reflects the different center of mass positions of the participating wave functions.

Returning to LL tunneling, we note a few important points. First, all LL resonances terminate on the right end at a specific bulk filling factor when magnetic depopulation removes the corresponding edge state from the sample, clearly seen for the black data in Fig. 3b. In particular, tunneling involving LL$_2$ with $n = 2$ is observed up to $B_z \approx 1.1$ T, terminating at the corresponding bulk filling factor $\nu = 6$, labeled on the top axes in Fig. 3b. Here, spin occupancy $g = 2$ because of a spin-unresolved case. The resonances for LL$_3$ with $n = 3$ are already lost above $B_z \approx 0.8$ at $\nu = 8$, independent of which LW mode is used as a spectrometer (compare red, black and light blue data in Fig. 3b).

**Spin splitting and Landau level depopulation.** A set of bright vertical features appears in the upper half of Fig. 3a (corresponding to the dashed vertical lines of integer filling factors in Fig. 3b), whose position is coincident with the disappearance of LL resonances. These features are even more visible in Supplementary Fig. 5. These result from probing the flat part of the LL dispersion i.e. they reflect the bulk filling factor, and account for the majority of the measured tunneling signal in Fig. 3 prior to differentiation of the data. For example at $B_z = 0.7$ T, shown in the level schematics of Fig. 2c, LL$_4$ is aligned with the chemical potential causing a resonance with the right Fermi-points of LW$_1$, LW$_2$, and LW$_3$ as indicated with black dots. While applying a positive in-plane magnetic field shifts the LW dispersions to the right and hence preserves resonant tunneling, this condition is lost for sufficiently negative $B_y$. Consequently, the vertical lines, corresponding to magnetic depopulation of a LL in the bulk, appear predominantly at positive $B_y$.

Beyond the vertical lines, the smooth evolution of LL tunneling resonances from Fig. 2 develops shoulder-like structures at larger $B_z$, clearly seen in Fig. 3d. The shoulders appear exactly at the transition between bulk filling factors (vertical lines in Fig. 3b) and are attributed to Fermi level pinning to LLs and impurity states, respectively, previously only accessible through

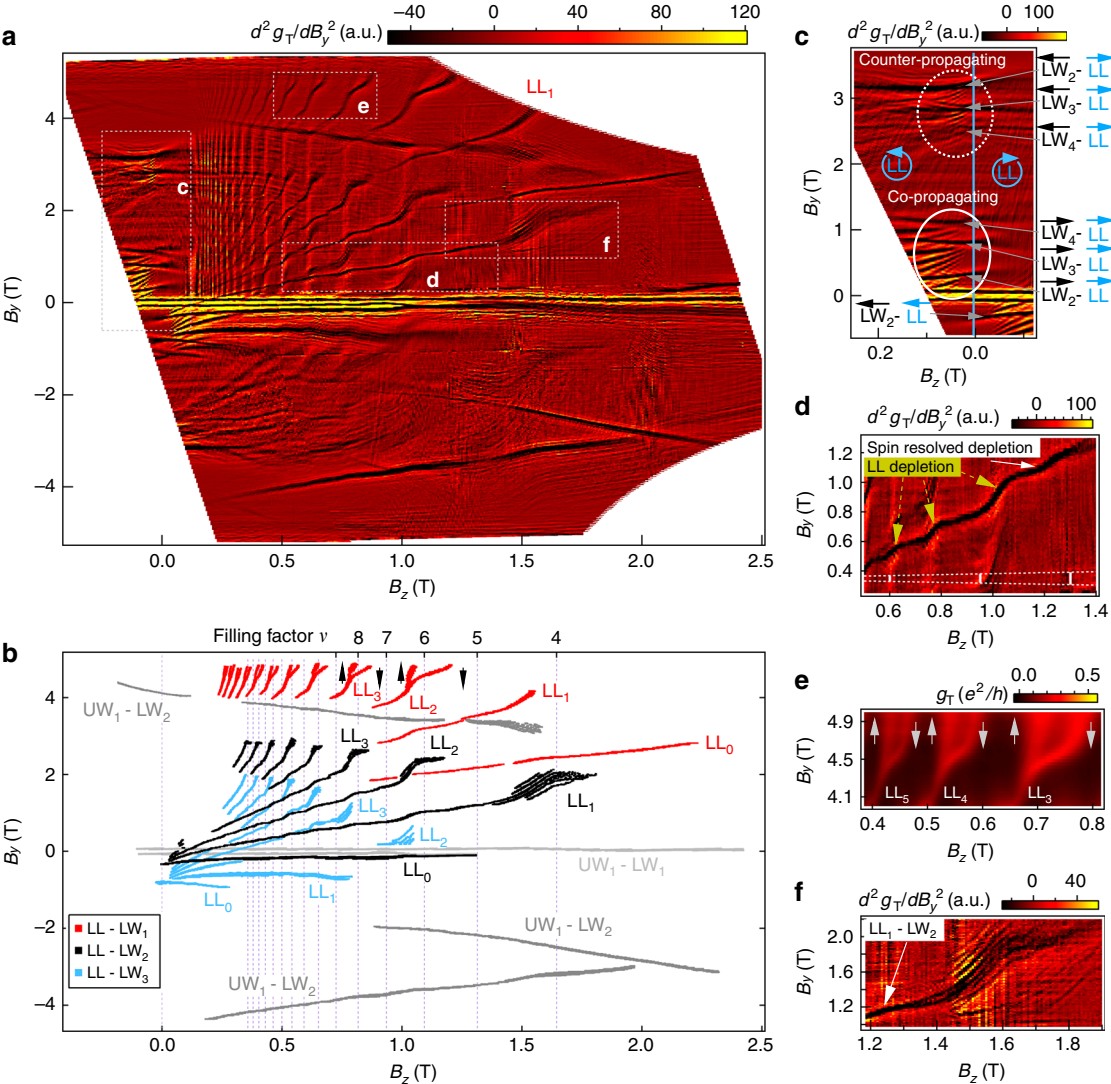

**Fig. 3** Magnetic depopulation and spin splitting of integer quantum Hall edge states. **a** Second derivative with respect to $B_y$ of the differential tunnel conductance $\left(d^2 g_T/dB_y^2\right)$ as a function of magnetic fields $B_y$ and $B_z$. **b** Extracted resonance positions from **a**. Red, black, and light blue data correspond to tunneling between edge modes and the first (LW$_1$), second (LW$_2$) and third (LW$_3$) lower wire mode. **c–f** Zoom-in of **a** for regions of interest: **c** Landau fans for $B_z < 0$ (counter-clockwise edge states ⇔ right moving edge state at cleaved edge), imaged with co-propagating (white solid ellipse) and counter-propagating wire modes (white dashed ellipse). **d** Jumps in the resonance position whenever the bulk filling changes. The three vertical bars of growing height indicate a distance of 2 nm in real space. The height $\Delta B_y$ of the bar is given by $\Delta B_y = \Delta Y B_z/d$, where $\Delta Y$ is the distance in real space. Thus, the real space resolution is improving with increasing magnetic field $B_z$. **e** LL spin splitting clearly visible even in undifferentiated raw data (tunneling conductance $g_T$). **f** Branching out of resonances at the transition to magnetic depopulation

investigation of the bulk 2DEG properties[42,43]. Here, we also note that the momentum resolution and the corresponding real space resolution of this spectroscopy technique improves with perpendicular magnetic field (white bars in Fig. 3d) and reaches the nanometer range for fields above 1 T. The resonance width depends on the degree to which momentum conservation is broken during tunneling, i.e. breaking of translational symmetry due to disorder and the finite size of the tunneling region. Finally, also, any variation of the tunneling distance between the upper and lower system, such as single-atomic steps in the growth plane or other crystal defects, will add to the observed broadening.

While each LL carries two spin-resolved sub-bands, energetically split by the total Zeeman energy given by magnetic fields $B_z$ and $B_y$, the corresponding difference in Fermi wave vectors is too small to be resolved by means of this spectroscopic method. However, at large in-plane magnetic fields, the interplay between Hartree term and exchange interactions[44] may lead to the

formation of spin-polarized strips where spin split sub-bands are also separated in real space[45]. As a consequence, tunneling resonances split up for the raw data in Fig. 3e and the spectroscopy becomes spin selective.

**Analytical model of resonant tunneling.** In the last part of this article, we develop an analytical model[19,20,24,32], and in addition provide numerical predictions (see Supplementary Note 3) for the evolution of LLs in the limit of hard wall confinement using a 1D single-particle Schrödinger solver. The perpendicular magnetic field introduces an additional local parabolic confinement, centered at each GC position $Y$, hence condensing bulk electrons into discrete LLs with well-known Hermite-Gaussian wave functions. We assume that upon approaching the hard wall, LLs remain at their bulk energy $E_n^{bulk} = \hbar \omega_c \left(n + \frac{1}{2}\right)$ until the tail of the wave function intersects with the hard wall ($Y \approx \sigma_n$ for LL$_n$, with $\sigma_n$ the half width of the LL wave function). When moving $Y$ even closer

to the edge or beyond, the hard wall retains the wave functions within the sample, thus separating in space the GC position $Y = k_x l_B^2$ and the wave function center of mass (CM) position, see Fig. 1a, b and 4a. As a consequence, LLs acquire kinetic energy and are simply lifted up the parabolic magnetic confinement until they cross the Fermi energy, thereby forming the conducting edge states (Fig. 1a, b and Supplementary Figs. 3, 4). Using these approximations, the LL dispersion $E_n(k_x)$ reads:

$$E_n(k_x) = E_n^{\text{bulk}} + \frac{\hbar^2}{2m^*}\Theta(\sigma_n - Y)\left(\frac{\sigma_n}{l_B^2} - k_x\right)^2, \qquad (1)$$

where $\Theta(x)$ is the Heaviside function. The condition for resonant tunneling is obtained by equating the LL spectrum at the Fermi energy with the lower wire dispersion $\epsilon_k^{(i)}$, shifted in $k_x$-direction to account for the momentum kick $eB_y d$ (tunneling to the lower system in presence of $B_y$) and $eB_z\Delta y_i$ (displacement $\Delta y_i$ of the $LW_i$ wave function CM with respect to the cleaved edge):

$$\epsilon_{k_x}^{(i)} = \frac{\left(\hbar k_x - eB_y d - eB_z\Delta y_i\right)^2}{2m^*} + \epsilon_0^{(i)}. \qquad (2)$$

Here, $\epsilon_0^{(l)}$ is an energy offset that accounts for the difference in band edges of 2DEG and respective lower wire mode with respect to the common Fermi energy. Combining Eqs. (1) and (2) we obtain the evolution of the tunneling resonances as a function of $B_y$ and $B_z$,

$$\frac{eB_y d}{\hbar} = \sqrt{\frac{2n+1}{l_B^2}} - \sqrt{k_{\text{F,2D}}^2 - \frac{2n+1}{l_B^2}} + \gamma_i - \frac{\Delta y_i}{l_B^2} \qquad (3)$$

where $\gamma_i = \sqrt{k_{\text{F,2D}}^2 - 2m^*\epsilon_0^{(l)} i\hbar^2}$ is a quantum wire mode dependent overall momentum shift.

## Discussion

Both the numerical and the analytical models capture the experimental tunneling resonances very well and result in very similar fitting parameters, shown in Fig. 4b, c for LL tunneling to $LW_2$ (black data from Fig. 3). We note that equally good fits are obtained for tunneling to other lower wire modes as well, using the same 2DEG density $n_{\text{2DEG}} = k_{\text{F,2D}}^2/2\pi$ and increasing quantum wire displacement $\Delta y_i$ for higher modes (see Supplementary Fig. 6), as expected—thus lending further support to the models. We emphasize that both models consistently deliver the CM positions, with similar nanometer precision as the GC positions extracted directly from the spectroscopy. This makes it possible to plot a full map of the magnetic field evolution of the edge states, see Fig. a. Throughout the process of increasing magnetic field Bz, the electron wave function is progressively compressed (from green to red curves). There are two stages of the edge state motion as magnetic field Bz increases: first, motion of the center of mass towards the hard wall (empty circles for $B_z < 2.78$ T) and motion away from the hard wall at larger fields, see also Supplementary Fig. 3. During the latter stage, the center of mass merges with the GC position (black and blue curves approach and then coincide for larger $B_z$ in Fig. 4a), followed by depopulation of the corresponding LL.

Despite the good match between experiment and non-interacting single-particle theory, there remain minor discrepancies. In particular, the shoulder structures at the transitions between integer bulk filling factors (Fig. 3d) and the spin splitting observed at large in-plane magnetic field (Fig. 3e) are not captured by the model. Furthermore, at the transition to magnetic depopulation, individual resonances are observed to branch out, clearly visible for, e.g., the LL$_1$–LW$_2$ transition in Fig. 3f. Splitting

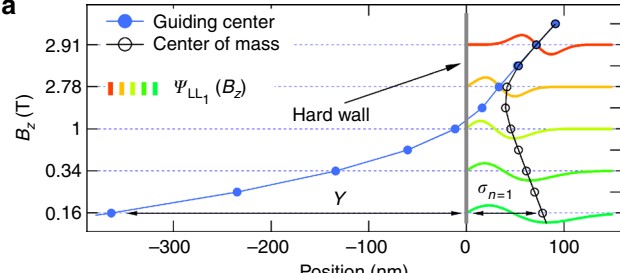

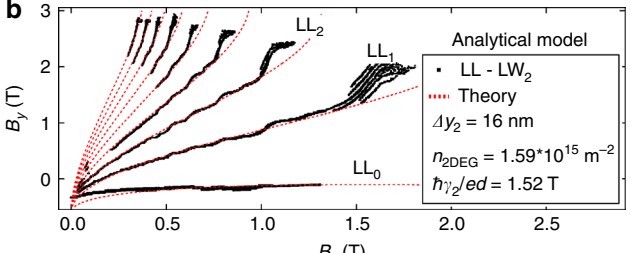

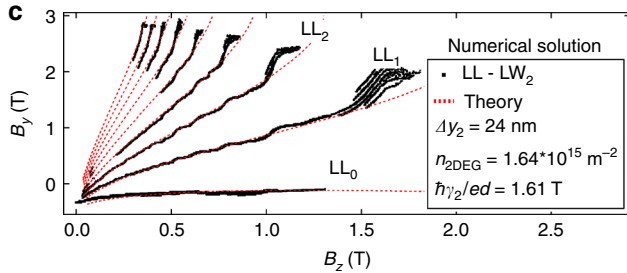

**Fig. 4** Comparison of experiment and theory. **a** Landau Level wave functions for particular values of $B_z$ chosen to visualize the important stages of magnetic field evolution. Note that the resulting vertical scale is highly nonlinear. The wave functions are obtained from a numerical Schrödinger solver, showing magnetic compression of the wave function and subsequent depopulation. The hard wall confinement completely separates wave function CM and GC position, the latter residing outside the physical sample for most of the B-field range. Hybridization of LLs and UW$_1$ would result in an additional node for LL wave functions at the Fermi energy. In **b**, **c** experimental data are compared to theoretical predictions from an analytical model and to numerical solutions from a single-particle Schrödinger solver, respectively

of single resonances could arise e.g. from edge reconstruction or may also result from the formation of stripe or bubble phases.

In summary, we employed momentum-resolved tunneling spectroscopy to image the evolution for the lowest ≈10 integer quantum Hall edge states of a GaAs 2DEG with nanometer resolution, and down to magnetic fields of $B_z \gtrsim 10$ mT ($\nu_{\text{bulk}} \approx 500$). We directly observe the chiral nature of integer quantum Hall edge states, as well as magnetic depopulation at the respective bulk filling factor. In addition, spin splitting is observed at the transition to depopulation. Theoretical predictions assuming the topologically gapped bulk spectrum and hard wall confinement reproduce very well the experimental data over the entire range of magnetic field, thus confirming the bulk to edge correspondence.

In the future, also fractional quantum Hall edge state can be investigated with the spectroscopy technique presented here. The present sample exhibits clearly visible $\nu = 4/3$ and $\nu = 5/3$ fractional states in conventional transport measurements. Imaging the fractional states by means of this highly sensitive momentum-resolved tunnel spectroscopy would be of great interest and can be addressed in future experiments. Fractional

states are stabilized by electron-electron interactions and are thus believed to exist only in the vicinity of the respective filling factor, in contrast to integer edge states that persist at all fields up to their magnetic depopulation. Because of power law exponents determining the tunneling conductance from the fractional quantum Hall edge states[46–50], a pronounced DC bias voltage dependence is expected for these states. This also makes it very interesting to explore another experimental knob, bias voltage, which controls the energy transfer during the tunneling event.

Beyond fractional states, the technique described here can also be applied to other topological insulator materials. In those systems where a wire exhibiting tunneling can be integrated or placed in parallel, edge states, both 1D or 2D in nature, can also be studied with this method with unprecedented resolution in a weakly invasive way. We note that ultra clean wires, such as used here to probe the edge states, are not necessary, in fact, thus opening the door to studying a variety of topological materials and their exotic edge states with this new tunneling spectroscopy.

## Methods

**Device layout.** The device used for this study is produced by means of the cleaved edge overgrowth method. It consists of a lower, 30 nm wide GaAs quantum well, separated by a 6-nm-thick AlGaAs tunnel barrier from the upper, 20-nm-thick GaAs quantum well[28]. A silicon doping layer above the upper quantum well provides free charge carriers, resulting in the formation of a 2DEG in the upper quantum well while the lower well remains unpopulated. The sample with pre-fabricated tungsten top gate is then cleaved inside the growth chamber and immediately overgrown on the sample edge (including a Si doping layer). Due to the additional side dopants, charge carriers are attracted to the sample edge, thereby forming strongly confined 1D channels (in upper and lower quantum well) along the entire cleaved edge. The 1D channels support a few (5 or less) transverse modes with sub-band spacing up to 20meV and mean free path exceeding 10μm[14]. Ohmic indium solder contacts to the 2DEG allow for transport studies. While the upper 1D system is well coupled to the 2DEG, the lower 1D channels are only weakly tunnel coupled, thus allowing for tunnel spectroscopy measurements.

**Measurements setup.** Tunneling spectroscopy measurements were recorded with the standard low frequency (5–10 Hz) lock-in technique with typically 6 μV AC excitation. All measurements were done at effectively zero DC bias using a specially designed low noise current preamplifier with active drift compensation (Basel Electronics Lab) ensuring $V_{DC} \lesssim 5$ μV.

Significant efforts were taken in order to obtain low electronic temperatures[51–57]. The present device is mounted on a home-built silver epoxy sample holder inside a heavily filtered dilution refrigerator with 5 mK base temperature. Roughly 1.5 m of thermocoax wire is used in combination with two stages of home-built silver epoxy microwave filters[53] to efficiently filter and thermalize each measurement lead, resulting in electronic sample temperatures around 10 mK.

**Numerical solution.** Numerical solutions are obtained by solving the 1D Schrödinger equation using Numerov's method. The hard wall confinement forces the electronic wave functions to be zero at that boundary. The perpendicular magnetic field gives an additional parabolic confinement. Its minimum is shifted away from the hard wall by the GC position $Y$. The energy of a given solution is then changed iteratively until a vanishing wave function at the hard wall is obtained.

**Data acquisition.** A vector magnet (8 T solenoid and 4 T split-pair) is used to provide the external magnetic field for spectroscopy measurements. Exceptional device stability is required in order to perform those extremely time-consuming B-field vs B-field maps in the main article. In particular, Fig. 3a is composed of 3 individual data sets with a total measurement time of roughly 6 weeks. In order to reduce the measurement time, here the magnetic field $B_z$ was scanned in a zig-zag fashion, i.e. taking data during ramping up and ramping down of $B_z$. However, due to the finite inductance of the magnet, a hysteretic behavior of the applied B-field results, which was accounted for by performing a non-linear correction to the measured data. The empty white spaces in Fig. 3a (round corners) are due to the accessible combined field range of the vector magnet. A slight sample misalignment with respect to the $y$- and $z$-direction is accounted for by tilting the experimental data in Figs. 2–4.

## Data availability

The data of all the Figures of the main manuscript and the relevant code are available on a Zenodo repository (https://doi.org/10.5281/zenodo.1251622).

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

## Acknowledgements

We thank Carlos Egues, Bertrand Halperin, Daniel Loss, and Jelena Klinovaja for very helpful discussions and we thank M. Steinacher and S. Martin and their teams for technical assistance. This work was supported by the Swiss Nanoscience Institute (SNI), NCCR QSIT, Swiss NSF, and the European Microkelvin Platform (EMP). A. Y. was supported by the NSF Grant No. DMR-1708688., The work at UCLA was supported by NSF under Grant No. DMR-1742928. The work at Princeton University was funded by the Gordon and Betty Moore Foundation through the EPiQS initiative Grant GBMF4420, and by the National Science Foundation MRSEC Grant DMR 1420541.

## Author contributions

T.P., C.P.S, D.M.Z conceived the experiment, analyzed the data and wrote the manuscript. T.P. and C.P.S. performed the experiments and numerical calculations. D.H. and Y.T. developed and carried out the theoretical work. G.B., A.Y., L.N.P., K.W.W. designed and fabricated the sample. All authors discussed the results and commented on the manuscript.

## Additional information

**Competing interests:** The authors declare no competing interests.

