## [Peer Review File · Nature Communications]

Reviewers' comments:

Reviewer #1 (Remarks to the Author):

I found this paper a joy to read. It describes in detail the plethora of experimental data, and presents a clear physical interpretation. The data are really striking - very strong signal of the edge states of the various Landau levels, with several copies due to tunneling from different wire states. The analysis and the interpretation explain most of the data, and overall the paper is an important contribution, which should be definitely published in Nature Physics. I have two minor questions, which the authors should clarify before publication: 1. The distance between the wire state and the edge states in the 2DEG is increasing with the Landau level index, as the edge states associated with these Landau levels are deeper in the 2DEG. I would expect that the signal would decrease exponentially with the LL index, but I see only weak dependence in the data. Can the authors deduce from the dependence of the signal on the LL index the distance of the different edge states from the edge of the system? 2. The authors simulate the 2DEG as having an sharp, infinite barrier. Indeed the cleaved edge may be emulated by such a potential, but, if I understand the structure correctly, then one of wires separate the 2DEG from that cleaved edge. Thus the density of electrons in that upper wire serve as an additional potential. Did the authors take this into account? Can one change the density in the upper wire so as to affect the actual boundary potential the electrons in the 2DEG see? Maybe this can be used to study actual edge reconstruction in the 2DEG.

Reviewer #2 (Remarks to the Author):

This paper reports evidence of momentum-resolved resonant tunneling between quantum Hall edge states and a deeply bound wire at the sharp edge of a cleaved-edge overgrown quantum Hall system. The experiment is challenging and the fact that the data is so convincing and clear is a testament to the quality of the work presented here. The results deserve publication in a prominent venue such as Nature Communications.

However, the manuscript has some drawbacks. The description tends to be a bit rambling, with some important aspects like defining the relevant quantum numbers being put into incidental remarks in the middle of a data description instead of being placed front and center in the problem description. Also key aspects of some of the conceptual plots are quantitatively inaccurate (Fig. 1) or highly confusing (Fig. 4a). At some points, the manuscript reads more like a thesis chapter than a well-honed journal article. The description is occasionally incomplete or underdefined, leaving confusing questions in the mind of the reader. A critical editing of the article for conciseness and clarity would improve the readability and therefore the likelihood that this work will be cited.

Several explicit points that require clarity are described below:

(1) Fig. 1: The Fig. 1a and 1b are helpful conceptually, but are quantitatively incorrect and must be corrected. These quantitative inaccuracies confuse the issue by creating physically impossible dispersion scenarios. (Figs. 1d, 1e and the insets of Fig2, etc. are all quantitatively accurate, on the other hand.) In Fig. 1a, b, for a simple hard-wall potential, the guiding center dispersion (dark blue line) must cross the energy $E = 1.5 \hbar \omega_c$ at guiding center coordinate $Y = 0$. This is because the ground state energy of a state that is bisected by a hard wall has a node at its guiding center position $Y = 0$, yielding a wavefunction at the wall that is identical to the first excited state in the bulk which is antisymmetric around $Y = 0$ and therefore also has a node in its center. The width of the ground-state bulk gaussian wavefunction in Fig. 1b is furthermore not accurate in the following manner. The width of the wavefunction (blue) for the ground state Landau level is far too narrow. The

wavefunction's energy in the guiding center dispersion (bold blue line) will only increase once the finite tail of the wavefunction (light blue gaussian) overlaps with the hard-wall confinement potential at $Y = 0$. As it is currently drawn, the guiding center dispersion starts to curve upward when the wavefunction is much too far away from the edge. As a rule of thumb, the dark blue and light blue dispersions in panel 1b MUST BE IDENTICAL TO THE CORRESPONDING DISPERSIONS IN PANEL 1a, but simply scaled in Y by the smaller magnetic length and scaled in E by the increased cyclotron energy. Thus the dispersion in 1b must ALSO cross $Y = 0$ at the energy $E = 1.5 \hbar \omega_c$. In all cases, it would be better if these curves and the corresponding wave functions were simply calculated rather than misrepresented by inaccurate hand-drawn approximations, but at the very least, respect for the correct magnetic length scale and energy scale must be preserved.

(2) P. 2, 2nd full paragraph: Regarding the statement in the last full paragraph on p. 2 about the hybridization of Landau-level edge states and deeply bound wire states, the authors would perhaps benefit from referring to the two references of Steinke, et al., below:

L. Steinke, P. Cantwell, E. Stach, D. Schuh, A. Fontcuberta i Morral, M. Bichler, G. Abstreiter, M. Grayson, "Hartree simulations of coupled quantum Hall edge states in corner-overgrown heterostructures," *Physical Review B* 87, 165428 (2013).

L. Steinke, P. Cantwell, D. Zakharov, E. Stach, N. J. Zaluzec, A. Fontcuberta i Morral, M. Bichler, G. Abstreiter, and M. Grayson, "Nanometer scale sharpness in corner overgrown heterostructures," *Applied Physics Letters* 193, 193117 (2008).

This work is extremely relevant to the present manuscript since it discusses wavefunctions and dispersions that involve a hybridization of a sharp quantum Hall edge and a deeply bound accumulation wire at the sharp edge. Steinke et al. shows that when a Landau-level edge state coexists with a deeply bound wire ground state, the exact solution to the quantum mechanical wavefunction is almost identical to taking the Landau-level edge state and projecting out the deeply bound wire state. This concept is exactly what the authors propose in this discussion, but Steinke et al have already conducted such an analysis for a very similar system and have proven that the wavefunctions so derived are extremely accurate. However, it is worth noting that Steinke et al also demonstrated that as the magnetic field strength increases, the anticrossing energy scale becomes quite large and the naive perturbative coupling implied in Fig. 1e is no longer valid, as anticrossing gaps become of order $\hbar \omega_c$. Such a limit is reached in the extreme quantum limit, as soon as the magnetic length starts to become as small as the triangular wire confinement length scale.

(3) On p. 3 within the first full paragraph, after the sentence, "only little momentum transfer and correspondingly small $|BY|$ is required to bring the modes into resonance", perhaps the clarifying comment or something similar would be in order: "And because the UW and LW states both share the same real space position in Y , the resonant tunneling condition is independent of B_z , thus generating a horizontal stripe in Fig. 2."

(4) Fig. 3: The colors in panel 3e appear to have inverted: red => black, black => red. This causes some confusion as all the other panels seem to be direct scaled zooms of the existing main panel 3a without any color adjustments.

(5) P. 4, 2nd to last full paragraph: "First, all LL resonances terminate..." note that the resonances terminate both at high field AND at low field. It is helpful to explicitly mention that the termination of interest here is the high-field limit so that people are looking at the right end of each resonance curve.

(6) Also, the sentence “Note that the LL index i denotes the orbital states counting from $i = 0$, while the filling factor includes the spin degeneracy” seems misplaced. The fundamental description of how the Landau levels and filling factors are indexed should be stated earlier in the definition of the system, not mentioned parenthetically as an aside in the middle of a data description. At the point where these terms are defined, there needs to be an explicit description:

$$\nu = 2*i + g$$

where $\nu = 1, 2, 3 \dots$ is the filling factor, $i = 0, 1, 2, \dots$ is the Landau level index, and $g = 1, 2$ is the spin occupancy – 1 for spin polarized Landau level, 2 for spin up + spin down both occupied. Some sort of mathematical definition like this needs to be explicitly stated so that the reader knows the relation between ν and i .

(7) Then in the paragraph on p. 4, 2nd to last full paragraph, here is where an explicit reference to the top axis, where the filling factor ν is plotted, needs to be made. Then one can mention explicitly how the filling factor is related to the Landau level index i , which is the subscript of the LL label. An explicit mention that the spin-unresolved case ($g = 2$ in the above formula), then can be described to pair up $i = 1$ to $\nu = 4$, and $i = 2$ to $\nu = 6$ and $i = 3$ to $\nu = 8$, as seen for the termination of the curves LL1, LL2 and LL3. The description as it stands is very sparse and confusing and it is very easy to lose the reader early if explicit care is not taken.

(8) P. 4, last full paragraph: “Second, a set of vertical lines appears in the upper half of Fig. 3a (dashed lines in Fig. 3b),” Very confusing. Are the authors referring to the dashed vertical lines that are labeled by the filling factor ν ? If so, why do the authors not simply refer to them as being the dashed lines that are labeled by the filling factor ν ? It is awkward to have this description of an experimental feature, and nowhere in the paragraph do you explicitly mention that these are indexed by the filling factor. The closest that the authors come is to say that the vertical lines “reflect the bulk filling factor”. Very obtuse choice of words.

(9) P. 5: “the real space resolution of this spectroscopy technique improves with perpendicular magnetic field (white bars in Fig. 3d)” – how is it an improvement? All three white scale bars are labeled 2 nm, so the resolution seems to be the same.

(10) Fig. 3d: The panel 3d is labeled with “LL depletion” and “subband depletion”. To my mind, “LL depletion” and “spin resolved depletion” would be much better adjectives to describe the difference between the two conditions here.

(11) P. 6: On the top of page 6, the lead sentence “In the last part of this article, we develop an analytical model inspired by the work of Halperin [9],...” This analytical model was already developed before by Huber, et al. [11, 12, 13] for quantum Hall edges and by Steinke, et al [Refs above] for quantum Hall edges coincident with a deeply bound edge wire. These works need to be referenced as well since the fundamental ideas of the model described by the authors were already laid out in these prior works.

(12) Fig. 4a: The vertical axis labeled B_z of Fig. 4a is confusing. The first interval is $0.34 - 0.16 = 0.18$. The next interval is $1.00 - 0.34 = 0.66$. The next interval is $2.78 - 1.00 = 1.78$, and the final interval is $2.91 - 2.78 = 0.13$. There is no logic to why these B_z fields are chosen and why they are plotted as being equally spaced, when the actual intervals in B_z that are being covered differ by over an order of magnitude.

(13) P. 6: Equation (1) is not rigorous, it is just a hand-waving approximation to the actual dispersion

that is correct in the limit form $k_x \gg 0$ and $k_x \ll 0$. There is nothing wrong with this approximation. But the word "approximation" is never stated in association with this Ansatz, making the argument misleading. The authors need to explain that Eq. (1) is just an useful analytical estimate of the dispersion, not a rigorous derivation.

Reviewer #3 (Remarks to the Author):

The paper is the first to show rather precise direct momentum resolved tunneling of edge and bulk states and, as this is an important first, I think the paper should be published in Nature Communications.

I think it would be helpful if the authors would make a couple of changes. First, there is a large prior body of work focusing on 1d-1d tunneling from Amir Yacoby's group (refs. 17-24). There should be a few sentences describing the relation of this work to the prior work and the difference between this work and the prior work from Yacoby's group. In this regard, it would be useful for me and the reader to understand what technical hurdles, if any, were overcome to do this work - why didn't Yacoby's group do this experiment 15 years ago? That said, the 1D-2D nature of the work here is clearly different from the prior work.

I don't like the use of the word "topological" in this paper. It will confuse readers in thinking that there is a quantum spin Hall effect in these samples or some such other physics rather than the physics of 2D edge states that was first described well before the development of "topological" theories of Kane and Male, etc. The paper is about old-fashioned integer quantum Hall edge states. While the authors may be technically right in calling them "topological", I think it is a new use of the word that will just confuse some readers.

The paper overall is well written and clearly explains what is going on. It would be nice to include in Fig. 1 a picture of the wavefunction of LLO trapped in the magnetic potential and with a hard wall to show the reader the definitions of guiding center and center of mass. There is an attempt to describe this in Fig. 3s of the supplement, but I think a simple intuitive picture in Fig. 1 would help many readers.

Part of the power of this spectroscopy is that it is all done at zero bias. There are no heating or lifetime effects. It might be useful to point this out. That said, I do wonder what happens as a function of energy. Have the authors attempted to look at what happens with applied DC voltage? Is there a magnetic field induced tunneling gap (or Luttinger behavior similar to what is seen with only one edge state occupied) for edge states similar to that in the bulk?

While the paper represents an important step forward, there aren't big surprises in it. Aside from DC voltage, do the authors see anything interesting with varying temperature? The exchange gaps will close at high temperature - that would be expected. But does anything happen between say, 10 mK and 100 mK? It seems that the authors worked very hard to get the samples very cold in this experiment, but we don't know if it matters at all. Identification of physics only appearing at very low temperatures would be an interesting addition to this paper.

The authors describe future experiments looking for fractional quantum Hall states, but one wonders why they haven't looked at higher B_z in this paper? The features at high B_z appear to be disappearing, and I'm wondering if this is the result of the development of the magnetic field induced tunneling gap suppressing the signal. Again, it would be very interesting to understand the nature of any gaps for

tunneling into edge states.

In short, I recommend publication of this paper in Nature Communications. The paper provides a new window into wire and edge states. One can see a strong agreement with theory and there impressive clarity in the data, showing things like the results of momentum boosts from both + and - k-states in the wire tunneling into the chiral 2D edge states. There is a lot of new detail here and a strong understanding of most of it.

RESPONSES TO REFEREE 1

Reviewer 1: I found this paper a joy to read. It describes in detail the plethora of experimental data, and presents a clear physical interpretation. The data are really striking - very strong signal of the edge states of the various Landau levels, with several copies of the edge states due to tunneling from different wire states. The analysis and the interpretation explain most of the data, and overall the paper is an important contribution, which should be definitely published in Nature Physics. I have two minor questions, which the authors should clarify before publication: 1. The distance between the wire state and the edge states in the 2DEG is increasing with the Landau level index, as the edge states associated with these Landau levels are deeper in the 2DEG. I would expect that the signal would decrease exponentially with the LL index, but I see only weak dependence in the data. Can the authors deduce from the dependence of the signal on the LL index the distance of the different edge states from the edge of the system?

Reply: We would like to thank the Referee for the positive comments on our work. The signal strength depends on the wave function overlap between the edge state and the wire mode. Because of the hard wall confinement, the lowest LL edge states are pulled towards the edge of the sample (see Fig. 4(a) of the main text or Figs. 3S(f), 4S of the supplement), in contrast to the case of soft confinement, which leads to the formation of compressible and incompressible strips. In the latter case, the edge states of different Landau levels are sitting in different places in space, whereas in the case of a hard wall, all the edge state wave functions start at the hard wall and interpenetrate each other. The wave function of the lower wire is very narrow compared to the wave function of the Landau level edge states at small magnetic fields and is localized close to the sample edge. This leads to the strong overlap between the wire mode and the last bump of different Landau level wave functions. As a result, only an algebraic decrease of the tunneling signal is observed. To deduce the position of the edge states from the tunneling signal strength, self-consistent calculations are required. Currently, such calculations are being worked out.

Rev1: 2. The authors simulate the 2DEG as having an sharp, infinite barrier. Indeed the cleaved edge may be emulated by such a potential, but, if I understand the structure correctly, then one of wires separate the 2DEG from that cleaved edge. Thus the density of electrons in that upper wire serve as an additional potential. Did the authors take this into account?

Reply: We agree the density of electrons in the wires does create an additional potential. This potential, however, compensates the approx. triangular potential close to the sample edge that results from the side dopants present in the CEO samples (introduced in the overgrowth process after sample cleaving in order to attract charge to the sample edge and form quantum wires). Thus, the remaining potential "after filling up the quantum wire states" can be well approximated with an approximately flat bottom potential, as also previously shown by Ref. 31 (revised manuscript).

An exact treatment of the full electrostatic problem (top + side dopants) results in the formation of hybrid states rather than separate Landau levels and quantum wire modes, shown in Fig.1(e) of the manuscript. The hybridization leads to opening of gaps that could be potentially interesting and would be visible at high magnetic fields using this type of spectroscopy. While we are currently working on self-consistent solutions for the problem (which goes beyond the scope of the current

work), we note that already the simple single particle picture gives a good quantitative account of the experimental data.

Rev1: Can one change the density in the upper wire so as to affect the actual boundary potential the electrons in the 2DEG see? Maybe this can be used to study actual edge reconstruction in the 2DEG.

Reply: It would be very interesting to see the evolution of the edge reconstruction as a function of confinement potential. Potentially this could be done if we had a range of samples with different doping levels and correspondingly different wire densities or a sample with a side gate where the density can be changed in-situ. Unfortunately, such samples are currently not available.

RESPONSES TO REVIEWER 2

Reviewer 2: This paper reports evidence of momentum-resolved resonant tunneling between quantum Hall edge states and a deeply bound wire at the sharp edge of a cleaved-edge overgrown quantum Hall system. The experiment is challenging and the fact that the data is so convincing and clear is a testament to the quality of the work presented here. The results deserve publication in a prominent venue such as Nature Communications.

However, the manuscript has some drawbacks. The description tends to be a bit rambling, with some important aspects like defining the relevant quantum numbers being put into incidental remarks in the middle of a data description instead of being placed front and center in the problem description. Also key aspects of some of the conceptual plots are quantitatively inaccurate (Fig. 1) or highly confusing (Fig. 4a). At some points, the manuscript reads more like a thesis chapter than a well-honed journal article. The description is occasionally incomplete or underdefined, leaving confusing questions in the mind of the reader. A critical editing of the article for conciseness and clarity would improve the readability and therefore the likelihood that this work will be cited.

Reply: We thank the referee for these points and acknowledge these problems in the manuscript. We have revised the manuscript and have tried to implement all suggested changes. In particular, we have moved forward and merged two paragraphs which are now paragraphs 2 and three (left column) on the first page, marked in blue in the revised manuscript. The paragraphs now begins:

“Previously, tunneling spectroscopy of cleaved edge overgrowth wires has established the system as one of the best realization of a 1D ballistic conductor...”

We have also added the definition of the filling factor early in the paper (first paragraph on page 2) as requested.

“Throughout the paper the filling factor is defined as $\nu = 2n + g$, where $n = 0, 1, 2, \dots$ is the orbital Landau level index, and g is the spin occupancy.”

Rev2: Several explicit points that require clarity are described below:
 (1) Fig. 1: The Fig. 1a and 1b are helpful conceptually, but are quantitatively incorrect and must be corrected. These quantitative inaccuracies confuse the issue by creating physically impossible dispersion scenarios. (Figs. 1d, 1e and the insets of Fig2, etc. are all quantitatively accurate, on the other hand.) In Fig. 1a, b, for a simple hard-wall potential, the guiding center dispersion (dark blue line) must cross the energy $E = 1.5 \hbar \omega_c$ at guiding center coordinate $Y = 0$. This is because the ground state energy of a state that is bisected by a hard wall has a node at its guiding center position $Y = 0$, yielding a wavefunction at the wall that is identical to the first excited state in the bulk which is antisymmetric around $Y = 0$ and therefore also has a node

in its center. The width of the ground-state bulk gaussian wavefunction in Fig. 1b is furthermore not accurate in the following manner. The width of the wavefunction (blue) for the ground state Landau level is far too narrow. The wavefunction's energy in the guiding center dispersion (bold blue line) will only increase once the finite tail of the wavefunction (light blue gaussian) overlaps with the hard-wall confinement potential at $Y = 0$. As it is currently drawn, the guiding center dispersion starts to curve upward when the wavefunction is much too far away from the edge. As a rule of thumb, the dark blue and light blue dispersions in panel 1b MUST BE IDENTICAL TO THE CORRESPONDING DISPERSIONS IN PANEL 1a, but simply scaled in Y by the smaller magnetic length and scaled in E by the increased cyclotron energy. Thus the dispersion in 1b must ALSO cross $Y = 0$ at the energy $E = 1.5 \hbar \omega_c$. In all cases, it would be better if these curves and the corresponding wave functions were simply calculated rather than misrepresented by inaccurate hand-drawn approximations, but at the very least, respect for the correct magnetic length scale and energy scale must be preserved.

Reply: Thank you for pointing this out. We were just making a qualitative figure and we have now changed it to be also quantitatively accurate. We have made all suggested changes in Fig.1a and Fig.1b: evolutions of the guiding center position are calculated directly instead of drawing them qualitatively. The new curve satisfies the important limit and goes through the point $E = 1.5 \hbar \omega_c$ at guiding center coordinate $Y = 0$ for the lowest Landau Level for both Fig.1a and Fig.1b. We have also numerically calculated the evolution of the center of mass and the wave functions instead of drawing them qualitatively. So the new figure respects both the magnetic length and energy scaling.

Rev2: (2) P. 2, 2nd full paragraph: Regarding the statement in the last full paragraph on p. 2 about the hybridization of Landau-level edge states and deeply bound wire states, the authors would perhaps benefit from referring to the two references of Steinke, et al., below:

L. Steinke, P. Cantwell, E. Stach, D. Schuh, A. Fontcuberta i Morral, M. Bichler, G. Abstreiter, M. Grayson, "Hartree simulations of coupled quantum Hall edge states in corner-overgrown heterostructures," *Physical Review B* 87, 165428 (2013).

L. Steinke, P. Cantwell, D. Zakharov, E. Stach, N. J. Zaluzec, A. Fontcuberta i Morral, M. Bichler, G. Abstreiter, and M. Grayson, "Nanometer scale sharpness in corner overgrown heterostructures," *Applied Physics Letters* 193, 193117 (2008).

This work is extremely relevant to the present manuscript since it discusses wavefunctions and dispersions that involve a hybridization of a sharp quantum Hall edge and a deeply bound accumulation wire at the sharp edge. Steinke et al. shows that when a Landau-level edge state coexists with a deeply bound wire ground state, the exact solution to the quantum mechanical wavefunction is almost identical to taking the Landau-level edge state and projecting out the deeply bound wire state. This concept is exactly what the authors propose in this discussion, but Steinke et al have already conducted such an analysis for a very similar system and have proven that the wavefunctions so derived are extremely accurate. However, it is worth noting that Steinke et al also demonstrated that as the magnetic field strength increases, the anticrossing energy scale becomes quite large and the naive perturbative coupling implied in Fig. 1e is no longer valid, as anticrossing gaps become of order $\hbar \omega_c$. Such a limit is reached in the extreme quantum limit, as soon as the magnetic length starts to become as small as the triangular wire confinement length scale.

Reply: Yes, thank you very much for this comment. We are now citing the first suggested paper:

Steinke, L. et al. "Hartree simulations of coupled quantum Hall edge states in corner-overgrown heterostructures," Phys. Rev. B 87, 165428 (2013).

(page 3, left column, first paragraph, citation [31]) as we found it relevant for our study. In fact, we also found another paper:

Grayson, M. et al. "Metallic and insulating states at a bent quantum hall junction," Phys. Rev. B 76, 201304 (2007).

(reference [30]) addressing similar issues several years earlier, so we are also citing this as well. The second paper suggested by the referee is predominantly about the characterization of 2DEG corner structures and seems not that relevant for our work and we have therefore decided not to cite it. We have also found another important paper which we haven't cited so far and we have added it to the citations ([18]):

MacDonald, A. H. & Streda, P. "Quantized Hall effect and edge currents," Phys. Rev. B 29, 1616 (1984).

"In the last part of this article, we develop an analytical model [17, 18, 22, 30] ..."

Rev2: (3) On p. 3 within the first full paragraph, after the sentence, "only little momentum transfer and correspondingly small $|BY|$ is required to bring the modes into resonance", perhaps the clarifying comment or something similar would be in order: "And because the UW and LW states both share the same real space position in Y, the resonant tunneling condition is independent of B_z , thus generating a horizontal stripe in Fig. 2."

Reply: Thank you for this comment; we have added a very similar sentence to the manuscript.

On page 3, in the first column, last paragraph, we have inserted:

"These resonances are independent of B_z because the Y coordinates of both UW₁ and LW₁ modes are very similar"

Rev2: (4) Fig. 3: The colors in panel 3e appear to have inverted: red => black, black => red. This causes some confusion as all the other panels seem to be direct scaled zooms of the existing main panel 3a without any color adjustments.

Reply: Thank you for pointing this out. Panel 3e shows undifferentiated raw data while panel 3a shows the second derivative of the raw data with respect to magnetic field B_y . So these panels show two different physical quantities and have color bars which can't be compared directly.

We changed the caption to highlight this fact:

"LL spin splitting clearly visible even in undifferentiated raw data (tunneling conductance g_T)."

Rev2: (5) P. 4, 2nd to last full paragraph: "First, all LL resonances terminate..." note that the resonances terminate both at high field AND at low field. It is helpful to explicitly mention that the termination of interest here is the high-field limit so that people are looking at the right end of each resonance curve.

Reply: Thank you for this comment. We changed the following sentence to include your suggestion:

"First, all LL resonances terminate on the right end at a specific bulk filling factor ..."

Rev2: (6) Also, the sentence "Note that the LL index i denotes the orbital states counting from $i = 0$, while the filling factor includes the spin degeneracy" seems misplaced. The fundamental description of how the Landau levels and filling factors are indexed should be stated earlier in the definition of the system, not mentioned parenthetically as an aside in the middle of a data description. At the point where these terms are defined, there needs to be an explicit description:

$$\nu = 2i + g$$

where $\nu = 1, 2, 3 \dots$ is the filling factor, $i = 0, 1, 2, \dots$ is the Landau level index, and $g = 1, 2$ is the spin occupancy – 1 for spin polarized Landau level, 2 for spin up + spin down both occupied. Some sort of mathematical definition like this needs to be explicitly stated so that the reader knows the relation between ν and i .

Reply: Thank you for your suggestion. We have added the following sentence to define the filling factor (page 2, left column, first paragraph):

“Throughout the paper, the filling factor is defined as $\nu = 2n + g$, where $n = 0, 1, 2, \dots$ is the orbital Landau level index, and g is the spin occupancy.”

Rev2: (7) Then in the paragraph on p. 4, 2nd to last full paragraph, here is where an explicit reference to the top axis, where the filling factor ν is plotted, needs to be made. Then one can mention explicitly how the filling factor is related to the Landau level index i , which is the subscript of the LL label. An explicit mention that the spin-unresolved case ($g = 2$ in the above formula), then can be described to pair up $i = 1$ to $\nu = 4$, and $i = 2$ to $\nu = 6$ and $i = 3$ to $\nu = 8$, as seen for the termination of the curves LL1, LL2 and LL3. The description as it stands is very sparse and confusing and it is very easy to lose the reader early if explicit care is not taken.

Reply: Thank you for the comment. We have taken it into account and modified the corresponding sentence:

“In particular, tunneling involving LL₂ with $n=2$ is observed up to $B_z \approx 1.1$ T, terminating at the corresponding bulk filling factor $\nu=6$, labeled on the top axes in Fig. 3b. Here, spin occupancy $g=2$ because both spins are populated.”

Rev2: (8) P. 4, last full paragraph: “Second, a set of vertical lines appears in the upper half of Fig. 3a (dashed lines in Fig. 3b),” Very confusing. Are the authors referring to the dashed vertical lines that are labeled by the filling factor ν ? If so, why do the authors not simply refer to them as being the dashed lines that are labeled by the filling factor ν ? It is awkward to have this description of an experimental feature, and nowhere in the paragraph do you explicitly mention that these are indexed by the filling factor. The closest that the authors come is to say that the vertical lines “reflect the bulk filling factor”. Very obtuse choice of words.

Reply: In this paragraph, we refer to the bright vertical features present only in the region of $B_y > 0$ and $B_z > 0$ in Fig. 3a. To make our point clearer, we have changed this paragraph and included the reference to the supplementary figure (undifferentiated raw data) where the corresponding features are even more visible:

“Second, a set of bright vertical features appears in the upper half of Fig. 3a (corresponding to the dashed vertical lines of integer filling factors in Fig. 3b), whose position is coincident with the disappearance of LL resonances. These features are even more visible in Supplementary Fig. 5.”

Rev2: (9) P. 5: “the real space resolution of this spectroscopy technique improves with perpendicular magnetic field (white bars in Fig. 3d)” – how is it an improvement? All three white scale bars are labeled 2 nm, so the resolution seems to be the same.

Reply: All three scale bars correspond to a real space displacement of 2nm, but the length of these scale bars are different and increases for larger magnetic field B_z , thus the sensitivity is increasing. We modified Fig. 3d and the caption to emphasize this.

“The three vertical bars of growing height indicate a distance of $2\lambda_{nm}$ in real space. The height ΔB_Y of the bar is given by $\Delta B_Y = \Delta Y B_Z / d$, where ΔY is the distance in real space. Thus, the real space resolution is improving with increasing magnetic field B_Z .”

Rev2: (10) Fig. 3d: The panel 3d is labeled with “LL depletion” and “subband depletion”. To my mind, “LL depletion” and “spin resolved depletion” would be much better adjectives to describe the difference between the two conditions here.

Reply: We agree and have changed “subband depletion” to “spin resolved depletion” in panel 3d.

Rev2: (12) Fig. 4a: The vertical axis labeled B_z of Fig. 4a is confusing. The first interval is $0.34 - 0.16 = 0.18$. The next interval is $1.00 - 0.34 = 0.66$. The next interval is $2.78 - 1.00 = 1.78$, and the final interval is $2.91 - 2.78 = 0.13$. There is no logic to why these B_z fields are chosen and why they are plotted as being equally spaced, when the actual intervals in B_z that are being covered differ by over an order of magnitude.

Reply: Thank you for this comment. Please note that the evolution of the wave function is highly nonlinear when the corresponding Landau Level is close to the Fermi level. We have added a small section to the last paragraph on page 6 to explain this behavior:

“Throughout the process of increasing magnetic field B_z , the electron wave function is progressively compressed (from green to red curves). There are two stages of the edge state motion as magnetic field B_z increases: first, motion of the center of mass towards the hard wall (empty circles for $B_Z < 2.78$, T) and motion away from the hard wall at larger fields. During the latter stage, the center of mass merges with the guiding center position (black and blue curves approach and then coincide for larger B_Z in Fig. ref{fig:4}a), followed by depopulation of the corresponding LL.”

We have also modified the caption of Fig.4a to emphasize this.

“Landau Level wave functions for particular values of B_Z chosen to visualize the important stages of magnetic field evolution. Note that the resulting vertical scale is highly nonlinear.”

Rev2: (13) P. 6: Equation (1) is not rigorous, it is just a hand-waving approximation to the actual dispersion that is correct in the limit form $k_x \gg 0$ and $k_x \ll 0$. There is nothing wrong with this approximation. But the word “approximation” is never stated in association with this Ansatz, making the argument misleading. The authors need to explain that Eq. (1) is just an useful analytical estimate of the dispersion, not a rigorous derivation.

Reply: Thank you for pointing this out. We have modified the sentence with the model description (page 6, first column, last paragraph):

“We assume that upon approaching the hard wall, LLs remain at their bulk energy ...”

We also added the word “approximations” before the formula (1):

“Using these approximations the LL dispersion ...”

RESPONSES TO REVIEWER 3

Reviewer 3: The paper is the first to show rather precise direct momentum resolved tunneling of edge and bulk states and, as this is an important first, I think the paper should be published in Nature Communications.

I think it would be helpful if the authors would make a couple of changes. First, there is a large prior body of work focusing on 1d-1d tunneling from Amir Yacoby’s group (refs. 17-24). There should be a few sentences describing the relation of this work to the prior work and the difference between this work and the prior work from Yacoby’s group. In this regard,

it would be useful for me and the reader to understand what technical hurdles, if any, were overcome to do this work - why didn't Yacoby's group do this experiment 15 years ago? That said, the 1D-2D nature of the work here is clearly different from the prior work.

Reply: Thank you for the comment. This work was only possible because of the ability to independently and smoothly controlling two orthogonal magnetic fields using a vector magnet. This allows us to form the quantum Hall edge states applying some value of a magnetic field B_Z perpendicular to the 2D electron gas and scan an in plane field B_Y to perform momentum resolved tunneling from states with different momentum in the upper system.

To make this point also clear in the paper we have changed the section of the first paragraph on page two "Such wires are among ..." and converted it into a new paragraph about all previous works done on similar samples (first page, first column, second paragraph). We also have added a sentence about the importance of having independent control over two orthogonal magnetic fields:

"Here we use a vector magnet to independently control two orthogonal magnetic fields: one to form quantum Hall edge states and another to perform tunneling spectroscopy."

Rev3: I don't like the use of the word "topological" in this paper. It will confuse readers in thinking that there is a quantum spin Hall effect in these samples or some such other physics rather than the physics of 2D edge states that was first described well before the development of "topological" theories of Kane and Male, etc. The paper is about old-fashioned integer quantum Hall edge states. While the authors may be technically right in calling them "topological", I think it is a new use of the word that will just confuse some readers.

Reply: Thank you for your comment. We have added a sentence (first page, second column, first paragraph) to clarify this:

"Note that in this work we are studying integer quantum Hall edge states and not the spin Hall effect or any other topological state. However, this technique is also applicable to the latter states."

Rev3: The paper overall is well written and clearly explains what is going on. It would be nice to include in Fig. 1 a picture of the wavefunction of LL0 trapped in the magnetic potential and with a hard wall to show the reader the definitions of guiding center and center of mass. There is an attempt to describe this in Fig. 3s of the supplement, but I think a simple intuitive picture in Fig. 1 would help many readers.

Reply: Thank you for your suggestion. We have added a magnetic confinement parabola to Fig.1b (gray dashed parabolas) for two guiding center positions and show the corresponding wave functions.

Rev3: Part of the power of this spectroscopy is that it is all done at zero bias. There are no heating or lifetime effects. It might be useful to point this out. That said, I do wonder what happens as a function of energy. Have the authors attempted to look at what happens with applied DC voltage? Is there a magnetic field induced tunneling gap (or Luttinger behavior similar to what is seen with only one edge state occupied) for edge states similar to that in the bulk?

Reply: Thank you for this important comment. The DC bias is an additional control knob which we haven't explored yet in great detail. We plan to do bias dependent measurements in future experiments. We have added a sentence (first page second column first paragraph):

"We emphasize that this spectroscopy is done at zero bias, thus eliminating heating or lifetime effects."

Rev3: While the paper represents an important step forward, there aren't big surprises in it. Aside from DC voltage, do the authors see anything interesting with varying temperature? The exchange gaps will close at high temperature - that would be expected. But does anything happen between say, 10 mK and 100 mK? It seems that the authors worked very hard to get the samples very cold in this experiment, but we don't know if it matters at all. Identification of physics only appearing at very low temperatures would be an interesting addition to this paper.

Reply: This new technique opens new avenues for the direct and precise study of edge state reconstruction, spin exchange and Fermi level pinning. For a substantial part of the experiment presented here, it turns out that temperature plays a minor role. For example, the data in Fig. 2 looks essentially the same when measured at 0.5 K. There is a little bit more smearing, but resonances for all the Landau level edge states are still visible (apart possibly from the very faint ones). Having low temperatures is essential to study more fragile states, such as fractional quantum Hall edge states, on which we would like to focus in the future. In addition, the present sample was used in an earlier study to investigate helical nuclear order in the quantum wires, which is fully developed only at temperatures below roughly 100 mK [15]. The present spectroscopy technique may also be used to investigate the associated helical gaps in the electronic spectrum of the quantum wires, either through intra wire tunneling or using the integer quantum Hall edge states as a sensor.

Rev3: The authors describe future experiments looking for fractional quantum Hall states, but one wonders why they haven't looked at higher Bz in this paper? The features at high Bz appear to be disappearing, and I'm wondering if this is the result of the development of the magnetic field induced tunneling gap suppressing the signal. Again, it would be very interesting to understand the nature of any gaps for tunneling into edge states.

Reply: We agree that it would be very interesting to study tunneling at higher values of Bz. Unfortunately, our vector magnet was not able to reach fractional filling factors in the bulk of the sample for the orientation of the sample used in this paper. It is also important to note, that tunneling from fractional states is predicted to be suppressed by the power law exponents around zero bias which can potentially make it difficult to observe fractional states without applying DC bias. We added a sentence about this in the second paragraph second column on page seven (see below). In addition, as mentioned in the answer to the previous question, the edges states might also serve as a spectrometer to investigate helical gaps [15] in the adjacent quantum wires

“Because of power law exponents determining the tunneling conductance from the fractional quantum Hall edge states [43-47], a pronounced DC bias voltage dependence is expected for these states. This also makes it very interesting to explore another experimental knob, bias voltage, which controls the energy transfer during the tunneling event.”

Rev3: In short, I recommend publication of this paper in Nature Communications. The paper provides a new window into wire and edge states. One can see a strong agreement with theory and there impressive clarity in the data, showing things like the results of momentum boosts from both + and - k-states in the wire tunneling into the chiral 2D edge states. There is a lot of new detail here and a strong understanding of most of it.

REVIEWERS' COMMENTS:

Reviewer #1 (Remarks to the Author):

I find the response of the authors to my and the other referees' comments satisfactory, and recommend publication of the paper without additional changes.

Reviewer #2 (Remarks to the Author):

The authors have respectfully considered all the editorial suggestions made by this referee. Hopefully the paper will have a broader readership because of it. In particular, I think the revised Fig. 1a and 1b are very robust and clear.

I wish to nonetheless draw attention to the overstatements in the opening of the abstract, which are simply not true. The first sentence sells this work as describing a "topological material", and the second sentence as revealing a previously "impossible to access full evolution of edge states." In fact, though the quantum Hall effect is a topological system, it is incorrect to call it a "topological material". This error should be corrected. It is also misleading to imply that no one has ever conducted spectroscopy of edge states given the cited work of Huber et al. as well as the excellent work by the group of Jurgen Weis at the Max Planck Institute of Stuttgart, where scanning capacitive and scanning force microscopy has revealed all about the disorder and bulk conductivity giving way to conducting edge states. To be fair to the work that has preceded this publication, it would be preferred if the statement of the importance of what is presented here be limited to what is truly new, rather than being too grandiose and overstepping the bounds of correctness and making statements which are not supported by the literature record.

The work presented here is worthy of publication in Nature Communications, and presents a new and original perspective on quantum Hall edges. It is not necessary to weaken this paper by pretending that other prior work along the same lines didn't exist.

Here are a few suggested references from the Stuttgart group which detail the evolution of states from the bulk to the edges, in case the authors are not familiar:

- 1) Current distribution and Hall potential landscape towards breakdown of the quantum Hall effect: a scanning force microscopy investigation, K. Panos, R.R. Gerhardts, J. Weis, and K. von Klitzing Physical Review Letters 113, 076804 (2014).
- 2) Current-induced asymmetries of incompressible strips in narrow quantum Hall systems, R. R. Gerhardts, K. Panos, and J. Weis New Journal of Physics 15, 073034 (2013)
- 3) Cryogenic scanning force microscopy of quantum Hall samples: Adiabatic transport originating in anisotropic depletion at contact interfaces, F. Dahlem, E. Ahlswede, J. Weis, and K. von Klitzing, Physical Review B 82, 121305 (2010).

If the authors can find a proper way of putting the novel aspects of their work with proper recognition of the excellent work that preceded them, I believe their paper will be more

welcomed by the community.

Note that Ref. 11 has only an author listed, no journal.

Reviewer #3 (Remarks to the Author):

I am satisfied with the changes that the authors have made, and I believe that this paper is now ready for publication in Nature Communications.

RESPONSES TO REVIEWER #1

Reviewer 1: I find the response of the authors to my and the other referees' comments satisfactory, and recommend publication of the paper without additional changes.

Reply: We thank the referee for the review process and are pleased that in his opinion all comments have been addressed properly.

RESPONSES TO REVIEWER #2

Reviewer 2: The authors have respectfully considered all the editorial suggestions made by this referee. Hopefully the paper will have a broader readership because of it. In particular, I think the revised Fig. 1a and 1b are very robust and clear.

I wish to nonetheless draw attention to the overstatements in the opening of the abstract, which are simply not true. The first sentence sells this work as describing a "topological material", and the second sentence as revealing a previously "impossible to access full evolution of edge states." In fact, though the quantum Hall effect is a topological system, it is incorrect to call it a "topological material". This error should be corrected.

Reply: We thank the referee for the positive comment about the revised Fig. 1a and 1b.

We agree that the quantum Hall effect is not a material but rather a system. We have replaced the expression "topological materials" with "topological systems" in the first sentence of the abstract in order to avoid confusions.

Rev2: It is also misleading to imply that no one has ever conducted spectroscopy of edge states given the cited work of Huber et al. as well as the excellent work by the group of Jurgen Weis at the Max Planck Institute of Stuttgart, where scanning capacitive and scanning force microscopy has revealed all about the disorder and bulk conductivity giving way to conducting edge states. To be fair to the work that has preceded this publication, it would be preferred if the statement of the importance of what is presented here be limited to what is truly new, rather than being too grandiose and overstepping the bounds of correctness and making statements which are not supported by the literature record. The work presented here is worthy of publication in Nature Communications, and presents a new and original perspective on quantum Hall edges. It is not necessary to weaken this paper by pretending that other prior work along the same lines didn't exist. Here are a few suggested references from the Stuttgart group which detail the evolution of states from the bulk to the edges, in case the authors are not familiar:

1) Current distribution and Hall potential landscape towards breakdown of the quantum Hall effect: a scanning force microscopy investigation, K. Panos, R.R. Gerhardts, J. Weis, and K. von Klitzing, Physical Review Letters 113, 076804 (2014).

2) Current-induced asymmetries of incompressible strips in narrow quantum Hall systems, R. R. Gerhardt, K. Panos, and J. Weis, *New Journal of Physics* 15, 073034 (2013)

3) Cryogenic scanning force microscopy of quantum Hall samples: Adiabatic transport originating in anisotropic depletion at contact interfaces, F. Dahlem, E. Ahlswede, J. Weis, and K. von Klitzing, *Physical Review B* 82, 121305 (2010).

Reply: Thank you for pointing this out. Of course it is not our intention to offend any author by under-emphasizing the importance of their previous works. We are aware of the excellent works by the group of Jurgen Weis which show very clearly how the bulk of the sample alternates between incompressible and compressible state (when a bulk Landau level is about to be magnetically depopulated) thus giving rise to conducting edge states. While we have already cited some of this work in our previous submission (J. Weis and K. v. Klitzing, *Metrology and microscopic picture of the integer quantum Hall effect*, *Phil. Trans. R. Soc. A* 369, 3954 (2011)), we acknowledged the importance of those works and have included in addition all three citations suggested by the referee (see Ref [12, 13] in the first paragraph of the manuscript and Ref [42] on page 4, left column, first sentence of the 3rd full paragraph).

Furthermore, in order not to understate previous works, we have removed the following sentence from the abstract "So far, it has been impossible to access the full evolution of edge states with critical parameters such as magnetic field due to poor resolution, remnant bulk conductivity, or disorder."

In addition, we have modified the last two sentences of the first paragraph in the manuscript in order to properly state what is truly new in the present manuscript while, at the same time, not underemphasizing previous works. We have removed:

However, it has not been possible to map the positions of edge states with high resolution, and large magnetic fields were required to discriminate among individual edge states. As a consequence, experiments so far are limited to low filling factors and were not able to track the evolution of edge states over a wide range of parameters.

And we have replaced it with:

However, moderate spatial resolution and the requirement of large magnetic fields for discriminating among individual edge states have limited existing experiments to low filling factors and prevented tracking the evolution of quantum hall edges all the way down to low fields.

Rev2: If the authors can find a proper way of putting the novel aspects of their work with proper recognition of the excellent work that preceded them, I believe their paper will be more welcomed by the community

Reply: Thank you for this comment. Clearly, it is our goal that the paper is welcomed by the community. We hope that changes (listed in detail in the answer to the previous comment) now clearly reflects what is new in our work and yet properly acknowledges the excellent work that preceded ours.

Rev2: Note that Ref. 11 has only an author listed, no journal.

Reply: Thank you for pointing this out. We have adjusted Ref. 11 and added the journal:

Peng, L. et al. Observation of topological states residing at step edges of WTe₂. *Nature Communications* 8, 659 (2017).

RESPONSES TO REVIEWER 3

Reviewer 3: I am satisfied with the changes that the authors have made, and I believe that this paper is now ready for publication in Nature Communications.

Reply: We thank the referee for the review process and are pleased that in his opinion all comments have been addressed properly.